# The Effectiveness of Different Doses of Iron Supplementation and the Prenatal Determinants of Maternal Iron Status in Pregnant Spanish Women: ECLIPSES Study

**DOI:** 10.3390/nu11102418

**Published:** 2019-10-10

**Authors:** Lucía Iglesias Vázquez, Victoria Arija, Núria Aranda, Estefanía Aparicio, Núria Serrat, Francesc Fargas, Francisca Ruiz, Meritxell Pallejà, Pilar Coronel, Mercedes Gimeno, Josep Basora

**Affiliations:** 1Department of Preventive Medicine and Public Health, Faculty of Medicine and Health Sciences, Universitat Rovira i Virgili, 43201 Reus, Spain; lucia.iglesias@urv.cat (L.I.V.); nuria.aranda@urv.cat (N.A.); estefania.aparicio@urv.cat (E.A.); 2Tarragona–Reus Research Support Unit, Jordi Gol University Institute for Primary Care Research, 43202 Tarragona, Spain; meritxell.palleja@urv.cat (M.P.); josep.basora@urv.cat (J.B.); 3Clinical Laboratory, University Hospital Joan XXIII, Institut Català de la Salut, Generalitat de Catalunya, 43005 Tarragona, Spain; nserrat.tarte.ics@gencat.cat; 4Sexual and Reproductive Health Service of Reus–Tarragona, Institut Català de la Salut, Generalitat de Catalunya, 43202 Tarragona, Spain; ffargas.tarte.ics@gencat.cat (F.F.); paqui.r.d@hotmail.com (F.R.); 5Meiji Pharma Spain S.A. (formerly Tedec-Meiji Farma S.A.) Alcalá de Henares, 28802 Madrid, Spain; p.coronel@tedecmeiji.com (P.C.); m.gimeno@tedecmeiji.com (M.G.); 6CIBERobn (Center for Biomedical Research in Physiopathology of Obesity and Nutrition), Instituto de Salud Carlos III, 28029 Madrid, Spain

**Keywords:** iron supplementation, pregnancy, randomized controlled trial, serum ferritin, hemoglobin, iron status, iron stores, *HFE* gene

## Abstract

Iron deficiency (ID), anemia, iron deficiency anemia (IDA) and excess iron (hemoconcentration) harm maternal–fetal health. We evaluated the effectiveness of different doses of iron supplementation adjusted for the initial levels of hemoglobin (Hb) on maternal iron status and described some associated prenatal determinants. The ECLIPSES study included 791 women, randomized into two groups: Stratum 1 (Hb = 110–130g/L, received 40 or 80mg iron daily) and Stratum 2 (Hb > 130g/L, received 20 or 40mg iron daily). Clinical, biochemical, and genetic information was collected during pregnancy, as were lifestyle and sociodemographic characteristics. In Stratum 1, using 80 mg/d instead of 40 mg/d protected against ID on week 36. Only women with ID on week 12 benefited from the protection against anemia and IDA by increasing Hb levels. In Stratum 2, using 20 mg/d instead of 40 mg/d reduced the risk of hemoconcentration in women with initial serum ferritin (SF) ≥ 15 μg/L, while 40 mg/d improved SF levels on week 36 in women with ID in early pregnancy. Mutations in the *HFE* gene increased the risk of hemoconcentration. Iron supplementation should be adjusted to early pregnancy levels of Hb and iron stores. Mutations of the *HFE* gene should be evaluated in women with high Hb levels in early pregnancy.

## 1. Introduction

Iron requirements increase during pregnancy. Since dietary sources cannot always prevent iron deficit, iron supplements are usually prescribed to women who plan to become pregnant. However, there is no consensus on the ideal iron dosage during pregnancy. Anemia is the most common and widespread nutritional disorder globally and a significant public health problem [1,2]. Anemia is attributed to iron deficiency (ID) in half of the cases in the general population [1,3] and in up to 90% of cases of pregnant women [4]. Studies show that an inadequate iron status during pregnancy can lead to adverse mother–child outcomes. In the mother, iron deficiency, anemia, and iron deficiency anemia (IDA) have been associated with preeclampsia, preterm delivery, and even miscarriage, and in the child with fetal growth restriction, low birth weight and impaired cognitive development [5,6,7,8,9,10]. Furthermore, some studies have underscored the importance of timing of ID and IDA, since some long–term consequences, especially regarding the development and functioning of the child’s brain, are irreversible, even after correcting iron levels [11,12]. As a result, it is essential to maintain good nutritional care even before getting pregnant, as well as throughout the whole gestation, to ensure an optimal health status for mother and baby.

In addition to participating as enzymatic cofactor in a wide range of metabolic reactions, iron is indispensable for the synthesis of hemoglobin (Hb), the synthesis and methylation of DNA, and oxygen transport [13,14]. The increase in blood volume and the formation of new tissue during pregnancy are the main mechanisms underlying the increased iron requirements [15,16,17]. Crucially, iron has a key role in neuronal proliferation, myelination, and the synthesis of several neurotransmitters during the development of the fetal brain [11,18]. Despite concerns about the state of prenatal iron, which caused the launch of public health policies to address iron deficiency [19], it is estimated that in Europe, around 25% of pregnant women become anemic during pregnancy [2,3,19]. The prevalence of ID is greater than the prevalence of IDA, and it often develops during the later months of pregnancy, even in women with sufficient iron stores at the start of the pregnancy [20]. In addition, while diet and supplementation are the main sources of iron, the maternal iron status is influenced by many other biological, lifestyle, and even social factors. According to published research, genetic alterations, ethnicity, obstetric history, toxic habits (i.e., smoking or alcohol), and socioeconomic status (SES) could have a defining role [21,22,23,24,25].

On the other hand, unnecessary or excessive iron supplementation might generate high levels of Hb, also known as the risk of hemoconcentration, in the second and third trimesters of pregnancy. This condition, which affects between 8.7% and 42% of pregnancies in industrialized countries [26,27], increases oxidative stress and blood viscosity, causing placental infarction and hindering the perfusion of oxygen and nutrients to the fetus [28,29,30,31]. Although hemoconcentration can be as harmful as iron deficiency for maternal health and children’s health, in clinical practice, iron supplementation is usually not adjusted to fit iron status.

The primary aim of this study was to evaluate the effectiveness of iron supplements during pregnancy in different doses adjusted to the Hb levels of the first trimester. As secondary outcomes, we described the percentage of ID, anemia, IDA, and risk of hemoconcentration in a large sample of pregnant Spanish women and the prenatal factors associated with maternal iron status at the end of pregnancy.

## 2. Materials and Methods 

### 2.1. Study Design

The ECLIPSES study [32] was a community randomized controlled trial (RCT) conducted in the province of Tarragona (Catalonia, Spain) between 2013 and 2017. The 791 participants were contacted in their primary care centers during the first routine visit with midwives and were included in the trial according to the following inclusion criteria: over 18 years of age, gestation time ≤12 weeks, no lab indication of anemia (Hb ≥ 110 g/L on week 12), ability to understand the official State languages (Spanish or Catalan), and the ability to understand the characteristics of the study. Women with multiple pregnancy, adverse obstetric history, those who had taken >10 mg iron daily during the three months prior to week 12 of gestation, and those who reported a previous severe illness (immunosuppression) or chronic disease that could affect their nutritional status (cancer, diabetes, malabsorption, or liver disease) were excluded. A signed informed consent was obtained from all participants.

The participants were allocated into two strata according their initial Hb levels on week 12 of pregnancy, as follows:

(1) Stratum 1: women with initial Hb levels between 110 and 130 g/L were prescribed 40 or 80 mg/d of iron supplementation.

(2) Stratum 2: women with initial Hb levels > 130 g/L were prescribed 40 or 20 mg/d iron supplementation. Although in clinical practice, only plasma Hb and serum ferritin (SF) levels are measured, we suspected that women with initial Hb > 130 g/L could have some alteration in the *HFE* gene which would predispose them to iron overload.

In addition to the recruitment visit before the 12th week of gestation, the study consisted of three visits throughout the pregnancy: at the 12th, 24th, and 36th weeks of gestation. Separately, the women attended routine pregnancy visits with their midwives and obstetricians.

During the first visit (12th week), the midwives delivered the supplements to the participants according to the intervention group to which they had been assigned. The prescription of each dose of supplements within the groups was randomized and triple blinded. The laboratories Tedec–Meiji made the same box for all different doses of supplements, so that the laboratory technicians, the clinical staff and the researchers did not know the dose of iron received by each woman until the study ended. Women were advised to take one pill per day until the next visit, at which time they had to return any left-over pills to evaluate adherence. An independent investigator compared the number of pills left over with the compliance reported by the participants. Good compliance was considered for women who reported having forgotten to take the supplement less than twice per week at every visit of the study. When they reported forgetting two or more times per week in any of the visits, compliance was considered low.

If women developed anemia in the middle of pregnancy (24th week), they received the usual treatment for anemia.

The sample size was calculated according to previous data from our research group [9,33], taking into account the risk of IDA and hemoconcentration during the third trimester of pregnancy as principal variables [32]. The study was designed in agreement with the Declaration of Helsinki/Tokyo. All procedures involving human subjects were approved by Clinical Research Ethics Committee of the Jordi Gol University Institute for Primary Care Research (*Institut d’ Investigació en Atenció Primària; IDIAP*), the Pere Virgili Health Research Institute (*Institut d’Investigació Sanitària Pere Virgili; IISPV*), and the Spanish Agency for Medicines and Medical Devices (*Agencia Española del Medicamento y Productos Sanitarios; AEMPS*). Signed, informed consent was obtained from all women participating in the study. This clinical trial was registered at www.clinicaltrialsregister.eu as EudraCT number 2012-005480-28 and at www.clinicaltrials.gov with identification number NCT03196882.

### 2.2. Data Collection

#### 2.2.1. Baseline Data (on Week 12 of Gestation)

Midwives and researchers of the study (dietitians) compiled clinical and obstetrical data from participants during the first visit. They obtained the following information during the personal interview and from specific questionnaires: date of birth, weight, height, blood pressure, parity (yes/no), number of previous children, planned pregnancy (yes/no), previous use of contraceptives (yes/no), and type of contraceptives. Medical and surgical history and obstetric data were also recorded.

Maternal age was classified as <25 years, 25–34 years, and ≥35 years. Each maternal pre–pregnancy body mass index (BMI, Kg/cm^2^) was categorized as underweight (BMI < 18.5), normal weight (BMI 18.5–24.9), overweight (BMI 25–29.9), or obese (BMI ≥ 30). 

The dietary assessment was obtained using a short food frequency questionnaire (FFQ) validated in our population [34] and filled by participants at each visit of the study. From this information, we were able to calculate the percentage of adherence to the Mediterranean diet [35], considered a high–quality dietary pattern. In addition, women were asked about their use of multivitamin supplements, including >10mg iron, which constituted an exclusion criteria for this study.

Lifestyle habits before conception were also recorded, including alcohol intake and smoking. To assess smoking, we used the Fagerström test [36] and women were classified as smokers and non-smokers at the first visit of the study. The International Physical Activity Questionnaire (IPAQ) [37] was used to record the physical activity (PA) of participants. They reported the time spent doing exercise of different intensity (vigorous, moderate or a walk lasting at least 10 minutes) during the previous week; the information was recorded as “days per week” in which physical activity of each intensity was performed, and the “hours” and “minutes” dedicated in each of those days. Women also reported amount of time spent sitting during a typical day. We used these data to calculate the metabolic equivalents of task.

Sociodemographic data of participants and their partners were also recorded. The educational level was classified into four groups: unfinished primary school (<12 years old), primary school (up to 12 years old), secondary school (up to 18 years old) and higher education, which included university and vocational studies. Regarding occupational status, women were classified as students, employed or unemployed. Women in employment were asked about their profession, which was classified following the Catalan Classification of Occupations (CCO-2011) [38]. All this information was used to calculate the family’s socioeconomic status (SES).

Regarding ethnicity, five categories were used: Caucasian, Latin American, Asian, Arab, and Black.

Blood samples were taken on week 12 of gestation to perform blood and genetics tests. Hematological parameters (Hb, mean corpuscular volume (MCV), and hematocrit) and some specific biochemical markers (serum ferritin (SF) and C–reactive protein (CRP)) were measured, and genetic mutations of the *HFE* gene (C282Y, H63D, and S65C) were checked for. The samples were stored in the BioBank for future use.

#### 2.2.2. Data Recorded during Scheduled Study Visits

Diet and physical activity were also evaluated at 24th and 36th weeks of gestation. In addition, blood was collected during both visits to analyze routine blood parameters, including Hb levels. On week 36, SF levels were also measured.

Any adverse effect from the supplementation was recorded and included in the statistical analyses.

#### 2.2.3. Definition of Iron Status

Anemia was defined as Hb < 110 g/L at 12th and 36th weeks and Hb < 105 g/L at 24th week of gestation. ID was defined as SF < 15 µg/L and IDA as anemia and one of the following criteria: SF < 15 µg/L or MCV < 70 fL. SF levels ≥ 15 µg/L was considered as non–deficient or normal iron stores.

### 2.3. Statistical Analysis

All statistical analyses were performed for the population by intention to treat (ITT) and per-protocol. The population by ITT considered all the participants that were initially included in the study; the per-protocol population, however, consisted only of those participants who complied with the protocol of the study. In the latter, therefore, we excluded women who developed anemia on visit 2, at 24 weeks of gestation. 

All analyses were performed separating the sample by stratum; i.e., according to the Hb levels in the first visit of the study. Student’s *t*-test and ANOVA were used to describe continuous variables (mean and SD), and the chi-squared test for categorical variables (percentages). Natural logarithm (Ln) transformation was applied to normalize the distribution of SF, increasing the validity of analyses, and using the median and interquartile ranges (IQR).

Multivariate regression models (multiple linear regressions and logistic regressions) were used to assess the effect of different doses of iron supplementation, along with other prenatal predictors, on maternal iron status on week 36 of pregnancy. The models were adjusted for the following variables: maternal age, parity, socioeconomic status, use of hormonal contraception prior to getting pregnant, planned pregnancy, smoking habit, alcohol intake, pre–pregnancy maternal BMI, gestational weight gain, Hb on week 12 of gestation, SF on week 12 of gestation, CRP on week 12 of gestation, *HFE* gene genotypes, maternal ethnic origin, physical activity as weekly mean of metabolic equivalent of task (METs), and adherence to Mediterranean diet. 

Furthermore, adjusted multivariate regression models were performed for each stratum, separating women with and without ID in the first trimester in order to explore whether iron supplementation acted differently according to iron reserves at the beginning of pregnancy. They were adjusted for the same variables previously mentioned, except for SF on week 12 of gestation. To avoid information overload, the tables only show the statistically significant regression models. 

SPSS (version 25.0 for Windows; SPSS Inc., Chicago, IL, USA) was used for statistical analyses. Statistical significance was set at *p* < 0.05.

## 3. Results

Of the total of 791 pregnant women included in the study at week 12 of pregnancy (529 from Stratum 1 and 262 from Stratum 2), the data shown in this article are based on the population by ITT, which consisted of of 534 women with data on week 36 (354 from Stratum 1 and 180 from Stratum 2). Attrition was due to: voluntary abandonment (22.75%); miscarriage (1.64%); emergence of exclusion criteria during pregnancy (5.82%), including serious or chronic illness that could affect the nutritional development (e.g., cancer, diabetes, and malabsorption); and participants lost to follow up (2.28%). Attrition was proportional in both Strata, as shown in the Flowchart (Figure 1). In the Appendix A, we also show the analyses for the per-protocol population, which excluded anemic women at 24th week of gestation (11.7% in Stratum 1 and 2.7% in Stratum 2).

Table 1 shows the biological, lifestyle, and sociodemographic characteristics of participants at baseline. Compared with Stratum 1, women from Stratum 2 had a statistically significant higher baseline weight (64.83 and 67.17 kg, respectively, *p* = 0.017) and pre–pregnancy BMI (24.66 and 25.82, respectively, *p* = 0.001), and had gained significantly less weight during gestation (11.11 and 9.69 kg, respectively, *p* = 0.030). These differences did not translate into a significant effect on maternal iron status in the multivariate analyses. Table 1 also shows a trend (*p* = 0.075) toward a higher percentage of women with previous pregnancies in Stratum 1 (62.3%) than in Stratum 2 (55.7%). No significant differences in baseline characteristics were detected between women who dropped out of the study and women who reached the end of the intervention (Appendix A). 

We excluded the S65C mutation in the *HFE* gene from the multivariate analyses because of its low prevalence in our sample. For the same reason, subjects who were homozygous and heterozygous for H63D, together with the combined heterozygote H63D/C282Y, were grouped as “carrier of the H63D mutation.” We compared, therefore, three categories of mutation of the *HFE* gene in the multivariate analyses: wild type (WT/WT), heterozygous for C282Y/WT, and carrier of the H63D mutation. A similar situation occurred with maternal ethnic origin: we excluded Asian and Black subjects from subsequent analyses due to the low representation in the studied population, and only three final categories were considered: Caucasian, Arab, and Latin American.

Since diet is expected to influence iron status, adherence to the Mediterranean diet was compared among the different study groups (Figure 2), but no significant differences were found. 

We also performed a bivariate analysis comparing the percentage of women with and without risk of hemoconcentration on week 36 of gestation based on their initial Hb levels and *HFE* genotypes. As shown in Figure 3, we found that the H63D mutation in the *HFE* gene was significantly more prevalent among women from Stratum 2 (initial Hb levels > 130 g/L) who developed iron overload, compared with women who completed the pregnancy without risk of hemoconcentration (41.4% and 19.8%, respectively, *p* = 0.045). Similar results were obtained regarding the S65C mutation, which was observed in 6.9% of women who showed risk of hemoconcentration at the end of gestation, compared to 0.8% of women with normal Hb levels in the last trimester (*p* = 0.031). On the other hand, women with wild type (WT) genotype, i.e., without mutations in the *HFE* gene, were significantly more prevalent in the group from Stratum 2 who finished the pregnancy without risk of excess iron, than among women with Hb levels above 130 g/L on week 36 of gestation (74.6% and 51.7%, respectively, *p* = 0.015). 

In Table 2 we describe and compare the blood tests results of women on weeks 12 and 36 of gestation among the intervention groups; a significant difference (*p* = 0.042) was observed in SF levels at week 36 between 80 and 40 mg/d iron in Stratum 1 (median: 17.19, IQR: 11.53, and median: 14.70, IQR: 9.37, respectively) in the non–adjusted bivariate analyses. Table 2 also shows that the prevalence of ID on week 36 was significantly higher (*p* = 0.012) in the group receiving 40 mg iron per day (51%) than in women receiving 80 mg daily (38.2%). No other significant differences were observed between groups regarding prevalence of various iron states, although the risk of hemoconcentration in the third trimester of pregnancy showed a tendency to be higher among women who received 40 mg daily of iron (24%) than those receiving 20 mg of iron per day (13.1%). The same results were obtained in the per–protocol population (Appendix A).

Multivariate analyses were performed to explore the effectiveness of iron dosages evaluated in Stratum 1 (80 mg/d and 40 mg/d) and Stratum 2 (40 mg/d and 20 mg/d), as well as the impact of several possible prenatal determinant factors. The results of the adjusted multivariate analyses for Stratum 1, summarized in Table 3, show that taking an iron supplement of 40 mg/d instead of 80 mg/d significantly reduced SF levels (*p* = 0.026) and doubled the risk of ID (*p* = 0.022) at the end of pregnancy. In contrast, the intervention with different doses of iron did not significantly change Hb levels (*p* = 0.718), the risk of anemia (*p* = 0.166), or IDA (*p* = 0.299). SF levels in early pregnancy were positively associated with Hb levels (β: 1.70; SE: 0.66; *p* = 0.010) and SF levels (β: 0.60, SE: 0.04, *p* < 0.001) in the third trimester. Additionally, maternal age 35 years and above increased SF in week 36 of pregnancy (β: 0.21; SE: 0.07; *p* = 0.002). Increasing early pregnancy levels of SF showed a protective effect against ID (OR: 0.29; 95%CI: 0.19–0.45; *p* < 0.001), anemia (OR: 0.54; 95%CI: 0.32–0.90; *p* = 0.018), and IDA (OR: 0.32; 95%CI: 0.17–0.59; *p* < 0.001). No differences were observed between the iron dosages evaluated in Stratum 1 in relation to the risk of hemoconcentration at week 36 of gestation after adjusting for possible confounders (*p* = 0.481). The adjusted multiple linear regression model for the risk of hemoconcentration was not statistically significant (*p* = 0.071). Moreover, in Stratum 1, when the regression models were performed separating women with and without ID on week 12 (Table 4), we observed that only in women with ID, the dose of 80 mg/d instead of 40 mg/d increased Hb levels in the third trimester (β: 8.81; SE: 2.40; *p* = 0.001), protecting women against anemia and IDA (OR: 0.03; 95%CI: 0.01–0.60; *p* = 0.021, for both cases).

Similarly, Table 5 shows the results of multivariate analyses performed after selecting women from Stratum 2. Adjusting for possible confounding factors, we found that a daily iron supplementation of 20 mg as opposed to 40 mg during pregnancy reduced the risk of hemoconcentration by 69% (*p* = 0.035) without increasing the risk of any iron deficit states studied at the end of pregnancy. Similarly to Stratum 1, higher SF levels on week 12 of gestation were positively correlated with SF levels (β: 0.42; SD: 0.06; *p* < 0.001) in the last months. Increasing SF levels in early pregnancy protected, therefore, against ID (OR: 0.36; 95%CI: 0.19–0.68; *p* = 0.002), anemia and IDA (OR: 0.26; 95%CI: 0.08–0.66; *p* = 0.023, for both cases). Furthermore, the analyses showed the effect of maternal age on iron status on week 36, with women under 25 years presenting reduced SF levels (β: –0.28; SE: 0.11; *p* = 0.013), and women 35 years and older at lower risk of ID (OR: 0.37; 95%CI: 0.16–0.91; *p* = 0.029) than women between 25 and 34 years of age. It was also found that the middle–high SES, compared with low SES, protected against anemia and IDA (OR: 0.06; 95%CI: 0.01–0.40; *p* = 0.003, for both cases) in women who started pregnancy with Hb levels above 130 g/L. Regarding iron overload, in addition to the aforementioned effect of the low iron dose, higher Hb levels early in pregnancy and being a carrier of the H63D mutation significantly increased Hb levels on week 36 (β: 0.72; SE: 0.16, and β: 3.93; SE: 1.74, respectively) and the risk of hemoconcentration (OR: 1.20; 95%CI: 1.08–1.33, and OR: 3.09; 95%CI: 1.10–8.71, respectively). When the multivariate analyses were applied to the sample of women from Stratum 2, categorized according their initial iron stores, we found that compared to 20 mg, 40 mg of iron per day increased SF on week 36 (β: 0.39; SE: 0.15; *p* = 0.014) only in women with iron deficiency, while 20 mg/d reduced the risk of hemoconcentration (OR: 0.25; 95%CI: 0.07–0.85; *p* = 0.027) in women with initial iron stores within the normal range (Table 6).

In the multivariate analyses of Stratum 2, the results for the per–protocol and for the ITT populations were the same (Appendix A); for Stratum 1, the regression models for Hb levels, anemia and IDA lost statistical significance when women who were anemic at mid–pregnancy were removed from the sample. However, the results about the effects on SF levels and ID were the same as for the ITT population (Appendix A). 

## 4. Discussion

Despite the wealth of research on prenatal iron supplementation, there is a lack of consensus on the optimal iron dosage in relation to the characteristics of each woman. Consequently, we were determined to investigate the effectiveness of different doses of iron supplementation on preventing iron deficiency and excess iron in the last trimester of gestation. To our knowledge, few publications address the interplay of early maternal iron status and the effect of prenatal iron supplementation [39]. 

Firstly, we observed that the prevalence of ID found in both strata of our study population (38.2%–69.70%) was in the range of the European estimates for pregnant women published in the most recent reports [2,3]; regarding the prevalence of anemia (8.3%–13%) and IDA (7.3%–11.9%), our results were considerably lower than the estimates of the same reports (24.5% and 35%, respectively). In relation to the risk of hemoconcentration, we observed that its prevalence (~13%) was similar to previous reports from Spain by Arija et al. [27] and within the wide range reported in European countries (8.7% to 42%) [26]. We should underscore that most research focuses on iron deficiency, and only few studies have described the prevalence of excess iron; consequently, the estimates on iron overload are less updated and not as established. As expected, we observed a significantly higher prevalence of risk of hemoconcentration in Stratum 2 (13.1% for 20 mg/d and 24% for 40 mg/d) than in Stratum 1 (6.8% for 80 mg/d and 7.9% for 40 mg/d) at the end of pregnancy. This difference supports our hypothesis that women with normal–high initial Hb levels were at greater risk of iron overload, possibly due to the persistent effect that genetic alterations in the *HFE* gene exert on iron levels [40,41]. Our results also show a higher prevalence of *HFE* gene mutations in women from Stratum 2 at risk of hemoconcentration on week 36, as opposed to the higher prevalence of the wild type genotype in women who finished the pregnancy without that risk (see Figure 2). This highlights the influence of the genetic alteration in the *HFE* gene on the risk of iron overload in women with initial Hb levels > 130 g/L. Moreover, within Stratum 2, we found that the percentage of women at risk of hemoconcentration on week 36 in the group of 20 mg of iron per day was fifty percent less than in the group receiving 40 mg daily (13.1% and 24%, respectively, *p* = 0.063), confirming our hypothesis that low iron doses are the best option in this case. 

To clarify the effectiveness of different doses of prenatal iron supplementation on maternal iron status, the multivariate analyses were adjusted for several associated variables, including obstetric, biological, and socioeconomic conditions, as well as *HFE* gene genotype and iron–related blood parameters. In this regard, in women from Stratum 1 who began the gestation with Hb levels between 110 and 130 g/L, we observed that a daily dosage of 80 mg iron, as opposed to 40 mg, improved SF levels (b: 0.12, *p* = 0.026) and protected against ID (OR: 0.55, *p* = 0.022) at the end of pregnancy. Furthermore, when we explored the effect of iron supplementation in women within Stratum 1 according their initial iron reserves, we found that the higher dose of iron (80 mg/d) reduced the risk of anemia and IDA (OR: 0.03 and *p* = 0.021, for both cases) during the last months of gestation in women with iron–deficiency (SF < 15 µg/L, 14.2%) at the start of the pregnancy. In contrast, no significant effect was observed in women with SF ≥ 15 µg/L on week 12. These results respond to the physiological regulation of intestinal iron absorption in accordance with iron reserves, by which the body strongly regulates iron absorption when stores are sufficient [42,43]. On the contrary, and in agreement with Milman et al. [44], we did not find additional effects of high doses of iron in women with correct iron reserves at the beginning of the study. We can conclude that the usual prescribed dose of 40 mg daily would be effective in women with optimal initial iron reserves, but not in women with iron deficiency in early pregnancy. 

On the other hand, in Stratum 2 (initial Hb levels >130 g/L), women who received a daily dosage of 20 mg iron, compared with the group that received 40 mg, reduced the risk of hemoconcentration in the third trimester (OR: 0.31, *p* = 0.035), without increasing the risk of iron deficit. In this case, we should underscore that the risk of iron overload trebles in carriers of the H63D mutation of the *HFE* gene (OR: 3.09, *p* = 0.033). Accordingly, we would advise to prescribe low doses of iron to women with normal–high Hb (>130 g/L) levels in early pregnancy. Interestingly, the baseline prevalence of ID was higher than expected in this group (13.4%); similarly to *Stratum* 1, the different doses produced different results regarding iron status, which varied in accordance with the initial iron stores. The protective effect of 20 mg iron per day against the risk of hemoconcentration (OR: 0.25, *p* = 0.027) was only observed in women with sufficient iron reserves in early pregnancy (SF ≥ 15 µg/L). 

Based on these findings, we emphasize that iron supplementation during pregnancy should be adapted to the initial iron status of each woman, assessed not only by Hb levels but also by SF levels, to prevent both iron deficiency and iron overload at the end of gestation. These conclusions are in agreement with the valuable contributions of Milman et al. [45,46], Casanueva et al. [47], and Peña-Rosas and Viteri [26], who advocate adapting prenatal iron supplementation in view that both iron deficit and hemoconcentration have been associated with negative effects on maternal–child health [5,6,7,8,9,10,28,29,30]. 

Generally, in clinical practice only Hb levels are measured to monitor maternal iron status during pregnancy. However, while detecting anemia, Hb levels fail to diagnose ID. Our results show that the effects of iron supplementation vary as a function of initial iron reserves, indicating the importance of detecting ID at the beginning of the gestation. We advocate for the routine measurement of SF levels during antenatal checks. We also underscore that mutations in the *HFE* gene should be studied in women with normal–high Hb levels at the beginning of pregnancy to avoid excessive iron supply. Indeed, in relation to this, it is known that there is a racial difference in the prevalence of alterations in the *HFE* gene, being greater in the populations of northern Europe than in the Mediterranean countries [24]. This adds even more weight to the premise that it is necessary to evaluate the individual characteristics of women to prescribe the most efficient prenatal iron supplementation in each case.

In this study, the multivariate analyses have also revealed some prenatal determinants of maternal iron status at the end of pregnancy. For instance, high SF levels on week 12 were associated with the increase of Hb and SF levels on week 36 in both strata, reducing the risk of all iron deficiency states: 71% and 64% lower risk of ID, 68% and 74% lower risk of IDA, and 46% and 74% lower risk of anemia for Stratum 1 and Stratum 2, respectively (see Table 3 and Table 4). The results show that SF levels on week 36 increased with maternal age, and in Stratum 2, maternal age was also linked to the risk of ID in the third trimester of gestation, although to our knowledge, the underlying mechanism of this association is not yet elucidated. We found a protective role of middle–high SES against anemia and IDA (OR: 0.06, *p* = 0.003, for both cases), specifically in women from Stratum 2. This finding coincides with previous reports that conclude that low–income status is a risk factor for iron deficiency, presenting as ID, anemia and IDA, especially in developing countries [23,48,49]. This observation stresses that a low SES might be associated with less healthy lifestyles and under-attendance to antenatal care [50,51]. Also in agreement with other studies [40,52,53], we found that the H63D mutation in the *HFE* gene increased Hb levels (b: 3.93, *p* = 0.025) and trebled the risk of hemoconcentration (OR: 3.09, *p* = 0.033) on week 36. It is well established that mutations in the *HFE* gene are highly prevalent in Caucasian populations and that they are linked to iron overload [3,54]. It has been suggested that *HFE* gene mutations increase intestinal iron absorption [41,55]. In our study, therefore, the results suggest that the presence of some mutation in the *HFE* gene would increase iron absorption in women with initial Hb levels >130mg/L. Unexpectedly, maternal iron status was not significantly associated with diet in the multivariate analyses in any strata. Similarly, comparative analyses, including adherence to the Mediterranean diet failed to show significant differences between different supplementation groups. This result suggests that the diet was very similar among all the women in the study. Finally, the trend for a higher percentage of parity in Stratum 1 (62.3%) than in Stratum 2 (55.7%) suggests that previous births could weaken the iron status of women at the beginning of pregnancy. Interestingly, in the multivariate analyses parity seemed to reduce by 74% the risk of hemoconcentration in women of Stratum 1, but the results in the regression model were not statistically significant (*p* = 0.071). 

Understanding that the prenatal iron supplementation has a different effect on maternal iron status at the end of pregnancy according to initial levels of Hb and SF could contribute to improving public health policies and to adapting clinical practices to the population groups at risk. Taking into consideration other associated prenatal determinants of maternal iron status can also improve antenatal care. In view of the evidence presented in this study, we emphasize firstly, the importance of full iron reserves before pregnancy, in preparation for the high cost of iron during gestation; and secondly, we recommend that clinicians adapt iron supplementation to the initial levels of Hb and iron reserves (see Figure 4). To assess the presence of genetic mutations in the *HFE* gene in women with normal–high Hb levels and full iron reserves at the beginning of pregnancy can help to reduce the risk of hemoconcentration in this group. 

### Strengths and Limitations

The main strengths of the current community RCT are the large sample size (*n* = 791) and the extensive data collection regarding sociodemographic conditions, clinical information, obstetric data, and lifestyle, including diet and physical activity. In addition, testing for *HFE* gene mutations has added valuable information on the effect of genetic variability on iron metabolism and on the possible impact of personalized iron supplementation. Methodologically, we were able to evaluate the progression of iron status by monitoring blood parameters at different stages of pregnancy. However, some limitations must be taken into account when interpreting the findings of this study. Firstly, the notable dropout rate, although this is not uncommon in community interventions such as ours, which require several visits. No woman dropped out due to gastrointestinal side effects, since we used ferrimanitol ovalbumin instead of ferrous sulfate in our study. Another limitation was the lack of SF measurements in the 24th week of pregnancy, which would have strengthened the results. Since women gave birth in hospitals, data on maternal iron status at delivery were not available for inclusion.

## 5. Conclusions

In conclusion, we advise routine monitoring of Hb and SF during antenatal check–ups. These tools can be used in clinical practice to prescribe the optimal dose of iron supplements, with the ultimate aim of achieving the best pregnancy outcomes. In addition, the study of mutations in the *HFE* gene in women with normal–high Hb levels at the beginning of pregnancy could reduce the risk of hemoconcentration. Further studies are needed to assess the effect of mutations in the *HFE* gene on the maternal iron status and its interplay with prenatal iron supplementation to determine if there is a real need to use supplements in these cases. Future studies should also assess whether, in addition to the benefits for pregnant women, the supplementation with different doses of iron have benefits for their children.

## Figures and Tables

**Figure 1 nutrients-11-02418-f001:**
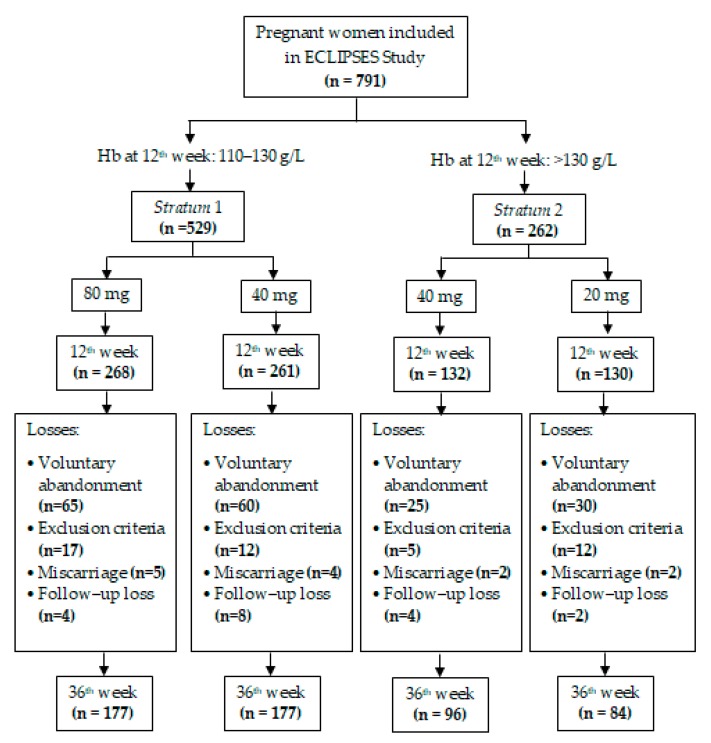
Flowchart of the study.

**Figure 2 nutrients-11-02418-f002:**
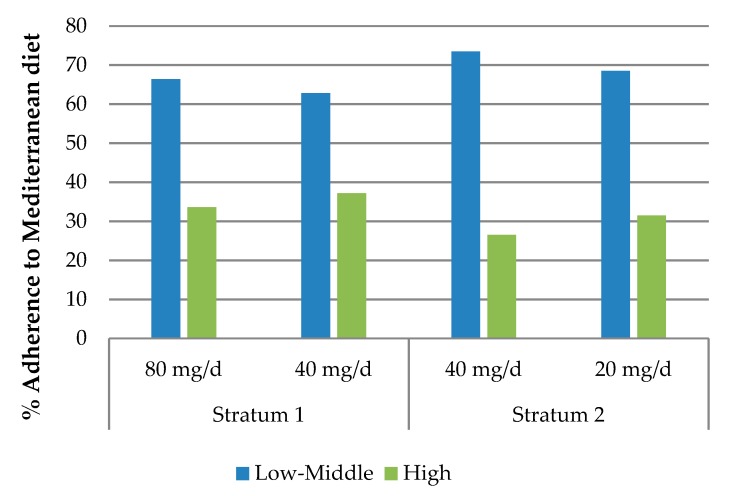
Adherence to the Mediterranean diet in the different groups of iron supplementation.

**Figure 3 nutrients-11-02418-f003:**
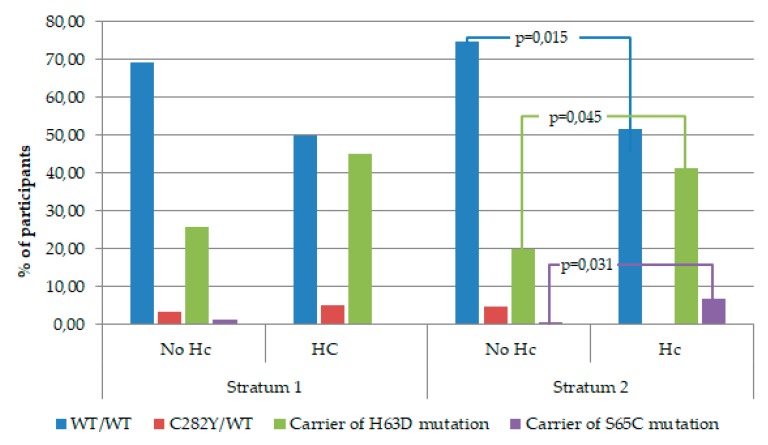
Percentage of women with and without risk of hemoconcentration (Hc) on week 36 of pregnancy, according to their initial hemoglobin (Hb) levels and *HFE* genotypes.

**Figure 4 nutrients-11-02418-f004:**
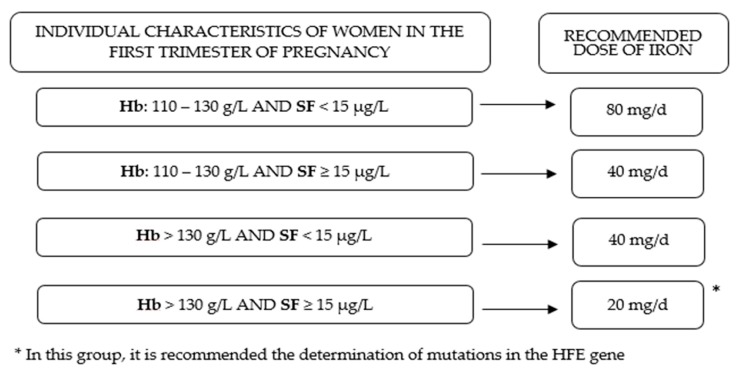
Adaptation of prenatal iron supplementation according the individual characteristics of women in the first trimester of pregnancy.

**Table 1 nutrients-11-02418-t001:** Baseline characteristics of the study population.

	*Stratum* 1 (*n* = 529)		*Stratum* 2 (*n* = 262)		*p*
	Mean	SD	Mean	SD	
Age, years	30.40	5.07	30.21	5.28	0.613
Weight, Kg	64.83	11.31	67.17	13.60	0.017
Pre–pregnancy BMI, Kg/m^2^	24.66	4.13	25.82	5.08	0.001
Gestational weight gain, Kg	11.11	8.17	9.69	9.47	0.030
	% (*n*)		% (*n*)		
Smoking	16.6 (88)		20.2 (53)		0.214
Parity	62.3 (329)		55.7 (146)		0.075
Planned pregnancy	79.8 (422)		80.5 (211)		0.801
Use of hormonal contraception	18.3 (97)		18.1 (47)		0.929
Pre–pregnancy BMI					
Underweight	1.3 (7)		2.3 (6)		0.314
Normal weight	60.9 (322)		51.5 (135)		0.012
Overweight	25.7 (136)		27.9 (73)		0.518
Obesity	12.1 (64)		18.3 (48)		0.018
*HFE* gene mutation	31.7 (130)		32.53 (68)		0.834
*HFE* genotype					
WT/WT	67.3 (280)		66.2 (141)		0.779
C282Y/WT	3.8 (16)		2.8 (6)		0.506
Carrier of H63D mutation	27.4 (114)		29.1 (62)		0.652
Carrier of S65C mutation	1.4 (6)		1.9 (4)		0.679
Family socioeconomic status					
Low	16.4 (87)		15.6 (41)		0.774
Middle	66.7 (353)		67.6 (177)		0.816
High	16.8 (89)		16.8 (44)		0.991
Maternal ethnic origin					
Caucasian	82.8 (405)		82.9 (203)		0.991
Asian	0.2 (1)		0.8 (2)		0.221
Arab	7.8 (38)		8.2 (20)		0.853
Black	1.8 (9)		0.4 (1)		0.114
Latin American	7.4 (36)		7.7 (19)		0.849
Adherence to Mediterranean diet					
Low–Middle	64.7 (342)		71.0 (186)		0.075
High	35.3 (187)		29.0 (76)		0.075

BMI: body mass index; WT: wild type. Sample size *HFE* genotype = 629; sample size maternal ethnic origin = 734.

**Table 2 nutrients-11-02418-t002:** Blood tests results of participants on week 36 of gestation according to supplementation dose.

	*Stratum* 1			*Stratum* 2		
	80 g/d	40 g/d	*p*	40 g/d	20 g/d	*p*
12th week						
Hemoglobin (g/L)	123.26 (5.32)	123.44 (4.77)	0.689	135.68 (4.59)	136.61 (4.44)	0.098
Serum ferritin (µg/L)	38.95 (26.10)	38.20 (25.05)	0.740	38.50 (28.98)	40.75 (30.00)	0.965
Mean corpuscular volume (fL)	87.08 (6.36)	87.30 (6.63)	0.696	88.53 (3.43)	88.54 (3.74)	0.980
C–reactive protein (mg/L)	0.73 (0.62)	0.74 (0.72)	0.815	0.72 (0.54)	0.70 (0.53)	0.779
Iron deficiency (%)	14.2 (38)	14.2 (37)	0.999	14.4 (19)	12.3 (16)	0.620
36th week						
Hemoglobin (g/L)	117.63 (7.55)	117.21 (8.35)	0.622	123.07 (10.19)	121.04 (8.85)	0.157
Serum ferritin (µg/L)	17.19 (11.53)	14.70 (9.38)	0.042	11.10 (8.10)	11.00 (6.80)	0.798
Mean corpuscular volume (fL)	89.31 (6.87)	88.19 (12.06)	0.261	90.23 (4.19)	89.61 (4.11)	0.299
C–reactive protein (mg/L)	0.76 (0.74)	0.71 (0.64)	0.470	0.70 (0.69)	0.75 (0.56)	0.593
Iron deficiency (%)	38.2 (71)	51 (98)	0.012	66 (68)	69.7 (62)	0.590
Iron deficiency anemia (%)	8.5 (15)	9.6 (17)	0.711	7.3 (7)	11.9 (10)	0.291
Anemia (%)	11.9 (21)	13 (23)	0.747	8.3 (8)	11.9 (10)	0.426
Hemoconcentration (%)	6.8 (12)	7.9 (14)	0.684	24 (23)	13.1 (11)	0.063

Continuous variables expressed as means (SD), except for serum ferritin, which is expressed as median (interquartile range). Categorical variables expressed in percentages (*n*).

**Table 3 nutrients-11-02418-t003:** The effects of the intervention with iron supplementation (40 or 80 mg/day) throughout pregnancy on hemoglobin and serum ferritin levels and on the risk of iron deficiency (ID), anemia, iron deficiency anemia (IDA), and hemoconcentration on the third trimester in women from Stratum 1.

**Hemoglobin levels**				
**Independent variables**	**β**	**SE**	***p***	**Model**
^a^ Intervention (0:80 mg/d, 1:40 mg/d)	−0.42	0.85	0.622	R^2^ = − 0.002 *p* = 0.622
^b^ Intervention (0:80 mg/d, 1:40 mg/d)	0.33	0.92	0.718	R^2^ = 0.031 *p* = 0.050
Hemoglobin on week 12 of pregnancy	0.25	0.10	0.015	
Serum ferritin on week 12 of pregnancy	1.70	0.66	0.010	
**Serum ferritin levels**				
**Independent variables**	**β**	**SE**	***p***	**Model**
^a^ Intervention (0:80 mg/d, 1:40 mg/d)	−0.10	0.06	0.085	R^2^ = 0.004 *p* = 0.085
^c^ Intervention (0:80 mg/d, 1:40 mg/d)	−0.12	0.05	0.026	R^2^ = 0.436 p< 0.001
Serum ferritin on week 12 of pregnancy	0.60	0.04	<0.001	
Maternal age (0:25–34 years, 1:<25 years)	0.05	0.08	0.559	
Maternal age (0:25–34 years, 1:≥35 years)	0.21	0.07	0.002	
**Iron deficiency (0:no, 1:yes)**				
**Independent variables**	**OR**	**95% CI**	***p***	**Model**
^a^ Intervention (0:80 mg/d, 1:40 mg/d)	1.69	1.12–2.54	0.012	R^2^ Nagelkerke = 0.022 *p* = 0.012
^c^ Intervention (0:80 mg/d, 1:40 mg/d)	1.82	1.09–3.03	0.022	R^2^ Nagelkerke = 0.241 *p* < 0.001
Serum ferritin on week 12 of pregnancy	0.29	0.19–0.45	<0.001	
**Anemia (0:no, 1:yes)**				
**Independent variables**	**OR**	**95% CI**	***p***	**Model**
^a^ Intervention (0:80 mg/d, 1:40 mg/d)	1.11	0.59–2.09	0.747	R^2^ Nagelkerke = 0.001 *p* = 0.747
^b^ Intervention (0:80 mg/d, 1:40 mg/d)	1.70	0.80–3.61	0.166	R^2^ Nagelkerke = 0.146 *p* = 0.027
Planned pregnancy (0:no, 1:yes)	3.57	1.00–12.80	0.050	
Serum ferritin on week 12 of pregnancy	0.54	0.32–0.90	0.018	
**Iron–deficiency anemia (0:no, 1:yes)**				
**Independent variables**	**OR**	**95% CI**	***p***	**Model**
^a^ Intervention (0:80 mg/d, 1:40 mg/d)	1.15	0.55–2.38	0.711	R^2^ Nagelkerke = 0.001 *p* = 0.711
^b^ Intervention (0:80 mg/d, 1:40 mg/d)	1.58	0.67–3.71	0.299	R^2^ Nagelkerke = 0.19 *p* = 0.004
Serum ferritin on week 12 of pregnancy	0.32	0.17–0.59	<0.001	
**Hemoconcentration (0:no, 1:yes)**				
**Independent variables**	**OR**	**95% CI**	***p***	**Model**
^a^ Intervention (0:80 mg/d, 1:40 mg/d)	1.18	0.53–2.63	0.684	R^2^ Nagelkerke = 0.001 *p* = 0.684
^b^ Intervention (0:80 mg/d, 1:40 mg/d)	1.44	0.52–3.97	0.481	R^2^ Nagelkerke = 0.166 *p* = 0.071
Genotype *HFE* (0:WT/WT, 1: carrier of H63D)	3.28	0.09–6.68	0.026	
Genotype *HFE* (0:WT/WT, 1: C282Y/WT)	1.93	0.21–18.02	0.566	
Parity (0:no, 1:yes)	0.26	0.09–0.76	0.014	

a Crude model. b Adjusted for: iron supplementation dosage, maternal age, use of hormonal contraception, pre–pregnancy maternal body mass index, gestational weight gain, *HFE* genotypes, maternal ethnic origin, hemoglobin on week 12, serum ferritin on week 12, c–reactive protein on week 12, socioeconomic status, weekly mean METS on week 12, smoking habit, alcohol intake, planned pregnancy, parity, mean caloric intake during pregnancy, and adherence to a Mediterranean diet. c Adjusted for: model b, except for hemoglobin on week 12.

**Table 4 nutrients-11-02418-t004:** The effect of the intervention with iron supplementation in Stratum 1 (0:80 mg/d, 1:40 mg/d) throughout pregnancy on maternal iron status on the third trimester, according to their initial iron stores.

**SF < 15 µg/L**				
**Hemoglobin levels**	**β**	**SE**	***p***	**Model**
Crude model	– 7.06	2.43	0.006	R^2^ = 0.129 *p* = 0.006
^a^ Adjusted model	– 8.81	2.40	0.001	R^2^ = 0.32 *p* = 0.003
**Iron deficiency (0:no, 1:yes)**	**OR**	**95% CI**	***p***	**Model**
Crude model	3.10	0.93–10.39	0.066	R^2^ Nagelkerke = 0.091 *p* = 0.060
^b^ Adjusted model	4.51	0.78–26.08	0.092	R^2^ Nagelkerke = 0.429 *p* = 0.013
**Anemia (0:no, 1:yes)**	**OR**	**95% CI**	***p***	**Model**
Crude model	5.50	1.05–28.75	0.043	R^2^ Nagelkerke = 0.145 *p* = 0.025
^a^ Adjusted model	29.14	1.67–508.56	0.021	R^2^ Nagelkerke = 0.596 *p* = 0.020
**Iron–deficiency anemia (0:no, 1:yes)**	**OR**	**95% CI**	***p***	**Model**
Crude model	5.50	1.05–28.75	0.043	R^2^ Nagelkerke = 0.145 *p* = 0.025
^a^ Adjusted model	29.14	1.67–508.56	0.021	R^2^ Nagelkerke = 0.596 *p* = 0.020
**SF** **≥** **15 µg/L**				
**Hemoglobin levels**	**β**	**SE**	***p***	**Model**
Crude model	0.75	0.88	0.395	R^2^ = −0.001 *p* = 0.395
^a^ Adjusted model	0.42	0.96	0.664	R^2^ = 0.035 *p* = 0.031

a Adjusted for: iron supplementation dosage, maternal age, use of hormonal contraception, pre–pregnancy maternal body mass index, gestational weight gain, *HFE* gene genotypes, maternal ethnic origin, hemoglobin on week 12, c–reactive protein on week 12, socioeconomic status, weekly mean of METS on week 12, smoking habit, alcohol intake, planned pregnancy, parity, mean caloric intake during pregnancy, and adherence to Mediterranean diet. b Adjusted for: model a, except for hemoglobin on week 12.

**Table 5 nutrients-11-02418-t005:** The effects of the intervention with iron supplementation (40 or 20 mg/day) throughout pregnancy on hemoglobin and serum ferritin levels and on the risk of ID, anemia, IDA, and hemoconcentration on the third trimester in women from Stratum 2.

**Hemoglobin levels**				
**Independent variables**	**β**	**SE**	***p***	**Model**
^a^ Intervention (0:40 mg/d, 1:20 mg/d)	−1.91	1.44	0.188	R^2^ = 0.004 *p* = 0.188
^b^ Intervention (0:40 mg/d, 1:20 mg/d)	−2.50	1.47	0.092	R^2^ = 0.116 *p* = 0.003
Genotype *HFE* (0:WT/WT, 1: carrier of H63D)	3.93	1.74	0.025	
Genotype *HFE* (0:WT/WT, 1: C282Y/WT)	1.34	3.70	0.718	
Hemoglobin on week 12 of pregnancy	0.72	0.16	<0.001	
**Serum ferritin levels**				
**Independent variables**	**β**	**SE**	***p***	**Model**
^a^ Intervention (0:40 mg/d, 1:20 mg/d)	0.02	0.07	0.734	R^2^ = −0.003 *p* = 0.734
^c^ Intervention (0:40 mg/d, 1:20 mg/d)	0.01	0.08	0.954	R^2^ = 0.218 *p* < 0.001
Maternal age (0:25–34 years, 1:<25 years)	−0.28	0.11	0.013	
Maternal age (0:25–34 years, 1:≥35 years)	0.12	0.10	0.221	
Serum ferritin on week 12 of pregnancy	0.42	0.06	<0.001	
**Low iron stores (0:no, 1:yes)**				
**Independent variables**	**OR**	**95% CI**	***p***	**Model**
^a^ Intervention (0:40 mg/d, 1:20 mg/d)	1.18	0.64–2.17	0.590	R^2^ Nagelkerke = 0.002 *p* = 0.590
^c^ Intervention (0:40 mg/d, 1:20 mg/d)	1.45	0.69–3.02	0.326	R^2^ Nagelkerke = 0.229 *p* = 0.003
Maternal age (0:25–34 years, 1:<25 years)	3.07	0.78–12.12	0.109	
Maternal age (0:25–34 years, 1:≥35 years)	0.37	0.16–0.91	0.029	
Serum ferritin on week 12 of pregnancy	0.36	0.19–0.68	0.002	
**Anemia (0:no, 1:yes)**				
**Independent variables**	**OR**	**95% CI**	***p***	**Model**
^a^ Intervention (0:40 mg/d, 1:20 mg/d)	1.48	0.56–3.96	0.428	R^2^ Nagelkerke = 0.007 *p* = 0.426
^b^ Intervention (0:40 mg/d, 1:20 mg/d)	2.01	0.44–9.09	0.364	R^2^ Nagelkerke = 0.468 *p* = 0.002
SES (0:low; 1:middle + high)	0.06	0.01–0.40	0.003	
Serum ferritin on week 12 of pregnancy	0.26	0.08–0.66	0.023	
**Iron–deficiency anemia (0:no, 1:yes)**				
**Independent variables**	**OR**	**95% CI**	**p**	**Model**
^a^ Intervention (0:40 mg/d, 1:20 mg/d)	1.78	0.62–4.74	0.295	R^2^ Nagelkerke = 0.013 *p* = 0.291
^b^ Intervention (0:40 mg/d, 1:20 mg/d)	2.01	0.44–9.09	0.364	R^2^ Nagelkerke = 0.468 *p* = 0.002
SES (0:low; 1:middle + high)	0.06	0.01–0.40	0.003	
Serum ferritin on week 12 of pregnancy	0.26	0.08–0.66	0.023	
**Hemoconcentration (0:no, 1:yes)**				
**Independent variables**	**OR**	**95% CI**	***p***	**Model**
^a^ Intervention (0:40 mg/d, 1:20 mg/d)	0.48	0.22–1.05	0.067	R^2^ Nagelkerke = 0.031 *p* = 0.060
^b^ Intervention (0:40 mg/d, 1:20 mg/d)	0.31	0.11–0.92	0.035	R^2^ Nagelkerke = 0.282 *p* = 0.004
Hemoglobin on week 12 of pregnancy	1.20	1.08–1.33	0.001	
Genotype *HFE* (0:WT/WT, 1: carrier of H63D)	3.09	1.10–8.71	0.033	
Genotype *HFE* (0:WT/WT, 1: C282Y/WT)	0.00	.	0.999	

a Crude model. b Adjusted for: iron supplementation dosage, maternal age, use of hormonal contraception, pre–pregnancy maternal body mass index, gestational weight gain, *HFE* gene genotypes, maternal ethnic origin, hemoglobin on week 12, serum ferritin on week 12, c–reactive protein on week 12, socioeconomic status, weekly mean of METS on week 12, smoking habit, alcohol intake, planned pregnancy, parity, mean caloric intake during pregnancy, and adherence to a Mediterranean diet. c Adjusted for: model b, except for hemoglobin on week 12.

**Table 6 nutrients-11-02418-t006:** The effect of the intervention with iron supplementation on Stratum 2 (0:40 mg/d, 1:20 mg/d) throughout pregnancy regarding maternal iron status in the third trimester, according to initial iron stores.

**SF** **<** **15 µg/L**				
**Serum ferritin levels**	**β**	**SE**	**p**	**Model**
Crude model	−0.24	0.14	0.083	R^2^ = 0.061 *p* = 0.083
^b^ Adjusted model	−0.39	0.15	0.014	R^2^ = 0.344 *p* = 0.021
**Iron deficiency (0:no, 1:yes)**	**OR**	**95% CI**	***p***	**Model**
Crude model	6.11	0.60–62.23	0.126	R^2^ Nagelkerke = 0.163 *p* = 0.085
^b^ Adjusted model	42.09	0.00–50.00	0.614	R^2^ Nagelkerke = 0.924 *p* = 0.001
**SF** **≥** **15 µg/L**				
**Hemoglobin levels**	**β**	**SE**	***p***	**Model**
Crude model	–1.75	1.54	0.260	R^2^ = 0.002 *p* = 0.260
^a^ Adjusted model	–2.00	1.56	0.200	R^2^ = 0.184 *p* < 0.001
**Serum ferritin levels**	**β**	**SE**	***p***	**Model**
Crude model	0.06	0.08	0.455	R^2^ = −0.002 *p* = 0.455
^b^ Adjusted model	0.08	0.09	0.368	R^2^ = 0.077 *p* = 0.008
**Anemia (0:no, 1:yes)**	**OR**	**95% CI**	***p***	**Model**
Crude model	1.34	0.43–4.20	0.611	R^2^ Nagelkerke = 0.004 *p* = 0.611
^a^ Adjusted model	1.02	0.20–5.10	0.984	R^2^ Nagelkerke = 0.383 *p* = 0.007
**Iron–deficiency anemia (0:no, 1:yes)**	**OR**	**95% CI**	***p***	**Model**
Crude model	1.63	0.50–5.39	0.421	R^2^ Nagelkerke = 0.010 *p* = 0.417
^a^ Adjusted model	1.02	0.20–5.10	0.984	R^2^ Nagelkerke = 0.383 *p* = 0.007
**Hemoconcentration (0:no, 1:yes)**	**OR**	**95% CI**	***p***	**Model**
Crude model	0.46	0.20–1.06	0.068	R^2^ Nagelkerke = 0.035 *p* = 0.061
^a^ Adjusted model	0.25	0.07–0.85	0.027	R^2^ Nagelkerke = 0.261 *p* = 0.001

a Adjusted for: iron supplementation dosage, maternal age, use of hormonal contraception, pre–pregnancy maternal body mass index, gestational weight gain, *HFE* gene genotypes, maternal ethnic origin, hemoglobin on week 12, C–reactive protein on week 12, socioeconomic status, weekly mean of METS on week 12, smoking habit, alcohol intake, planned pregnancy, parity, mean caloric intake during pregnancy, and adherence to a Mediterranean diet. b Adjusted for: model a, except for hemoglobin on week 12.

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
