# Peer review of "The Effectiveness of Different Doses of Iron Supplementation and the Prenatal Determinants of Maternal Iron Status in Pregnant Spanish Women: ECLIPSES Study"

_nutrients, 2019, doi:10.3390/nu11102418_

Round 1

Reviewer 1 Report

The authors have included the comments from previous version and have included additional analysis that has improved the manuscript. Also, the authors have edited the English language and style of the paper, that has made the manuscript easier to read.

There are minor comments for the revised manuscript:

1) Please include relevant references for the sentence in the introduction section: "Anemia is the most common and widespread nutritional disorder globally and a significant public health problem."

2) Line 75: "The primary aim of this study" instead of the "The primary outcome of the study".

3) Line 106: "triple blinded" or "tripe blinded"?

Author Response

The authors have included the comments from previous version and have included additional analysis that has improved the manuscript. Also, the authors have edited the English language and style of the paper that has made the manuscript easier to read.

There are minor comments for the revised manuscript:

1) Please include relevant references for the sentence in the introduction section: "Anemia is the most common and widespread nutritional disorder globally and a significant public health problem."

Two references from the WHO have been included regarding this sentence.

2) Line 75: "The primary aim of this study" instead of the "The primary outcome of the study".

The word “outcome” has been changed for “aim”.

3) Line 106: "triple blinded" or "tripe blinded"?

The typo has been fixed.

Reviewer 2 Report

The authors have addressed the previously formulated concerns, although i do not find a point by point rebuttal to the reviewer report. 
The authors should include a paragraph of strengths and weaknesses of the study in the discussion.

Author Response

The authors have addressed the previously formulated concerns, although i do not find a point by point rebuttal to the reviewer report.

The authors should include a paragraph of strengths and weaknesses of the study in the discussion.

We have included a paragraph of “Strengths and limitations” in the end of the discussion, before the conclusion.

Reviewer 3 Report

This report correctly focuses upon iron therapy during pregnancy both risks and harms over iron overload., which they postulate to be ferritin  > 15ug/l.  They provide guidelines for therapy to be personally based on HFE genetics to avoid iron overload based on ferritin and CRP levels.  

I suggest expansion of discussion references to include prior considerations of normal optimal ferritin levels e g 80-100 ug/l in the literature. Discussion of substantive HFE testing might be briefly expanded to include racial differences between Mediterranean and Northern European populations.

A well done and much need study and a well considered report

Author Response

This report correctly focuses upon iron therapy during pregnancy both risks and harms over iron overload, which they postulate to be ferritin > 15ug/l.  They provide guidelines for therapy to be personally based on HFE genetics to avoid iron overload based on ferritin and CRP levels. 

We would like to clarify in this regards that we define iron overload as hemoglobin > 130 mg/L, iron deficiency as serum ferritin levels <15 µg/L and SF levels ≥15µg/L as non–deficient or normal iron stores. The underscore sentence has been included in the manuscript (line 177) to clarify this definition.

After the results obtained by separating the sample according to the initial serum ferritin levels, we tried to highlight the increased risk of excess iron in those women who have correct iron stores and an initial risk due to high hemoglobin levels.

I suggest expansion of discussion references to include prior considerations of normal optimal ferritin levels e g 80-100 ug/l in the literature.

Although the normal values for serum ferritin has been for years established in different ranges, the WHO currently defines iron deficiency in adults as serum ferritin <15 µg/L (WHO, 2011). We followed this indication to define iron deficiency and, as opposed, serum ferritin ≥15 µg/L as normal or non-deficient iron stores. Although the optimal levels are higher, around 30-100 µg/L (Cullis, 2018), our aim in this work was only to discern between iron deficiency and non-deficiency.

World Health Organization. Serum ferritin concentrations for the assessment of iron status and iron deficiency in populations. Vitamin and Mineral Nutrition Information System. Geneva, World Health Organization, 2011 (WHO/NMH/NHD/MNM/11.2. Available from http://www.who.int/vmnis/indicators/serum_ferritin.pdf=.

Cullis, J. O., Fitzsimons, E. J., Griffiths, W. J., Tsochatzis, E. , Thomas, D. W. and , (2018), Investigation and management of a raised serum ferritin. Br J Haematol, 181: 331-340. doi:10.1111/bjh.15166

Discussion of substantive HFE testing might be briefly expanded to include racial differences between Mediterranean and Northern European populations.

A well done and much need study and a well-considered report

The following paragraph has been included in relation with the racial differences in terms of HFE alterations in the lines 389–393: “Indeed, in relation to this, it is known that there is a racial difference in the prevalence of alterations in the HFE gene, being greater in the populations of northern Europe than in the Mediterranean countries [24]. This adds even more weight to the premise that it is necessary to evaluate the individual characteristics of women to prescribe the most efficient prenatal iron supplementation in each case”.
